# In Vivo Imaging and Kinetic Modeling of Novel Glycogen Synthase Kinase-3 Radiotracers [^11^C]OCM-44 and [^18^F]OCM-50 in Non-Human Primates

**DOI:** 10.3390/ph16020194

**Published:** 2023-01-28

**Authors:** Kelly Smart, Ming-Qiang Zheng, Daniel Holden, Zachary Felchner, Li Zhang, Yanjiang Han, Jim Ropchan, Richard E. Carson, Neil Vasdev, Yiyun Huang

**Affiliations:** 1Brain Health Imaging Centre, Centre for Addiction and Mental Health, 250 College St., Toronto, ON M5T 1R8, Canada; 2Department of Psychiatry, University of Toronto, 250 College St., Toronto, ON M5T 1R8, Canada; 3Yale PET Center, Yale School of Medicine, 801 Howard Ave., New Haven, CT 06519, USA; 4Nanfang Hospital, Southern Medical University, 1838 Guangzhou Blvd North, Guangzhou 510515, China

**Keywords:** positron emission tomography, radiopharmaceutical, glycogen synthase kinase 3, non-human primates

## Abstract

Glycogen synthase kinase 3 (GSK-3) is a potential therapeutic target for a range of neurodegenerative and psychiatric disorders. The goal of this work was to evaluate two leading GSK-3 positron emission tomography (PET) radioligands, [^11^C]OCM-44 and [^18^F]OCM-50, in non-human primates to assess their potential for clinical translation. A total of nine PET scans were performed with the two radiotracers using arterial blood sampling in adult rhesus macaques. Brain regional time-activity curves were extracted and fitted with one- and two-tissue compartment models using metabolite-corrected arterial input functions. Target selectivity was assessed after pre-administration of the GSK-3 inhibitor PF-04802367 (PF-367, 0.03–0.25 mg/kg). Both radiotracers showed good brain uptake and distribution throughout grey matter. [^11^C]OCM-44 had a free fraction in the plasma of 3% at baseline and was metabolized quickly. The [^11^C]OCM-44 volume of distribution (*V*_T_) values in the brain increased with time; *V*_T_ values from models fitted to truncated 60-min scan data were 1.4–2.9 mL/cm^3^ across brain regions. The plasma free fraction was 0.6% for [^18^F]OCM-50 and *V*_T_ values (120-min) were 0.39–0.87 mL/cm^3^ in grey matter regions. After correcting for plasma free fraction increases during blocking scans, reductions in regional *V*_T_ indicated >80% target occupancy by 0.1 mg/kg of PF-367 for both radiotracers, supporting target selectivity in vivo. [^11^C]OCM-44 and [^18^F]OCM-50 warrant further evaluation as radioligands for imaging GSK-3 in the brain, though radio-metabolite accumulation may confound image analysis.

## 1. Introduction

The serine/threonine kinase glycogen synthase kinase 3 (GSK-3) is ubiquitous and abundant across the central nervous system. It is expressed in two isoforms, GSK-3⍺ and GSK-3β, of which GSK-3β is more widely studied. With over 100 substrates, GSK-3 is involved in regulating a multitude of functions including cellular metabolism, neuronal proliferation and synaptogenesis, and neurotransmitter signaling cascades [1,2,3,4]. GSK-3⍺/β dysregulation has been observed in a range of disorders including cancer, inflammatory diseases, and psychiatric disorders such as bipolar disorder, with GSK-3 indicated as the major target of lithium [3]. In neurodegenerative disorders, GSK-3 dysfunction has been linked to Parkinson’s disease, Huntington’s disease, and is implicated at several stages of disease progression in Alzheimer’s disease (AD). Its involvement in tau phosphorylation has led to the “GSK-3 hypothesis of AD” [5,6,7]. Accordingly, GSK-3 inhibitors are under ongoing investigations as potential therapeutics, but lack of target selectivity and undesirable side effects have presented significant challenges in clinical trials to date [3,6,8,9,10].

A GSK-3-selective positron emission tomography (PET) radiotracer suitable for brain imaging in clinical research could speed drug development by providing a method to confirm in vivo target engagement and refine dosing strategies, and facilitate disease diagnosis, monitoring, and patient selection for clinical trials. We reported the first GSK-3 candidate radioligand, [^11^C]AR-A014418, in 2005 [11], and a number of candidate radiotracers have since been developed by several groups and evaluated in preclinical models; however, none has yet advanced to evaluation in humans [12,13,14,15,16,17]. Common challenges are inadequate selectivity, poor brain penetrance, and a low in vivo binding signal. We recently reported [^11^C]PF-04802367 ([^11^C]PF-367) and derivatives as second-generation GSK-3 tracers with high selectivity and affinity for GSK-3α/β in vitro [13,15,18]. Of these, [^11^C]OCM-44 and [^18^F]OCM-50 (Figure 1) show the most promise with high selectivity and affinity (GSK-3β *IC*_50_, 2.5 nM and 0.8 nM, respectively) [18,19]. Initial in vivo PET scans in rats and non-human primates with both radiotracers showed good brain penetrance and fast clearance [19].

Given the promising results from in vitro and preliminary in vivo imaging studies of [^11^C]OCM-44 and [^18^F]OCM-50, the goal of this work was to perform full kinetic evaluation of these radiotracers in non-human primates to assess their potential for translation for human PET imaging studies. PET scans were performed in rhesus macaques with arterial blood sampling and quantitative kinetic modeling. Blocking studies with the GSK-3 selective antagonist PF-367 were performed to assess target selectivity and nonspecific binding. 

## 2. Results

### 2.1. Radiochemistry and Injection Parameters

[^11^C]OCM-44 and [^18^F]OCM-50 were produced at an average collected yield of 1285.4 ± 954.5 MBq with molar activity of 404.7 ± 107.1 GBq/µmol (*n* = 5) and 2066.9 ± 114.3 MBq with molar activity of 721.3 ± 334.0 GBq/µmol (*n* = 4), respectively. Radiochemical and chemical purity for each radiotracer was >90%. Average injection dose was 178 ± 2.1 MBq for [^11^C]OCM-44 and 164 ± 6.8 MBq for [^18^F]OCM-50. Injection parameters for baseline and blocking scans are shown in Table 1.

### 2.2. Blood Measurements

Plasma free fraction (*f*_p_) in baseline scans was 3.1% and 3.2% for [^11^C]OCM-44, and 0.6% in both scans for [^18^F]OCM-50. Metabolism was relatively fast for [^11^C]OCM-44, with <20% parent tracer remaining at 30 min post-injection, and somewhat slower for [^18^F]OCM-50, with approx. 40% parent remaining at 30 min (Figure 2, top). HPLC analysis for [^11^C]OCM-44 (Figure 2, bottom left) showed emergence of hydrophilic and lipophilic radio-metabolites throughout the scan duration, including some that were not distinctly separable from the parent by 60–90 min post-injection. The chromatogram for [^18^F]OCM-50 (Figure 2, bottom right) showed a distinct parent peak through the scan duration, with two distinct radio-metabolite peaks emerging well separated from the parent compound with significantly faster elution times.

### 2.3. Brain Distribution and Kinetics

Brain uptake of [^11^C]OCM-44 was moderate across grey matter regions and lower in white matter (Figure 3, middle left). Uptake of [^18^F]OCM-50 showed the same regional pattern but was lower overall (Figure 3, middle right). Time-activity curves (TACs; Figure 4) showed rapid uptake and washout of each radiotracer. [^11^C]OCM-44 reached peak tissue standardized uptake values (SUV) of 2–3 in grey matter, with these values decreasing to <1 across the brain by 30 min post-injection. For [^18^F]OCM-50, the peak SUV in brain was 1–2 with a similarly rapid decrease. 

### 2.4. Kinetic Modeling

For [^11^C]OCM-44, model fits were generally poor with the one-tissue compartment model (1TCM). Best fits based on Akaike Information Criterion (AIC) values were achieved using the two-tissue compartment model (2TCM) with assumed vascular fraction in brain of 5%. Given the potential of brain-penetrant radio-metabolites identified in HPLC analyses, models were fitted to data truncated to scan lengths between 30 and 105 min. In these analyses, volume of distribution (*V*_T_) values showed a linear increase with scan time across ROIs (Appendix A), such that regional *V*_T_ was 10–20% lower across cortical regions in one animal and 20–50% lower in the second when only the first 60 min of data were analyzed compared to the full 120 min. This is consistent with the accumulation of brain-penetrant metabolites across scan duration (Figure 2). To minimize the potential effects of such metabolites, analysis was performed using data from shorter scan periods. Because grey matter *V*_T_ estimates using less than 60 min of data showed higher uncertainty, *V*_T_ values from 2TCM fits to the first 60 min of data are presented in Table 2 and were used for subsequent comparisons and analysis of the blocking studies. 2TCM produced good model fits to 60-min data in baseline scans (Figure 4, left). Resulting *V*_T_ values ranged from 1.5–2.9 mL/cm^3^ (relative standard error [rSE] < 20% in most regions). Tissue influx rate constant (*K*_1_) values ranged from 0.18 mL∙cm^−3^∙min^−1^ in pallidum to 0.47 mL∙cm^−3^∙min^−1^ in putamen (Appendix A). 

For [^18^F]OCM-50, both the 1TCM and the 2TCM produced acceptable model fits across most regions in the baseline scans, with 1TCM preferred based on AIC values. Regional *V*_T_ values were lower than those of [^11^C]OCM-44 (<0.9 mL/cm^3^; Table 2) and did not show the same pattern of increase with scan duration (Appendix A). *K*_1_ ranged from 0.083 mL∙cm^−3^∙min^−1^ in amygdala to 0.21 mL∙cm^−3^∙min^−1^ in occipital cortex.

[^11^C]OCM-44 *V*_T_ values were highest in cingulate, frontal cortex, and other cortical regions, similar or slightly lower in striatum and hippocampus, and lowest in amygdala and pons. [^18^F]OCM-50 *V*_T_ was highest in frontal cortex but lowest in insula and caudate. In the Guo plot, only a weak linear relationship was observed between regional *V*_T_ values of the two radiotracers (*n* = 1; Appendix A).

### 2.5. Blocking Studies

PF-367 was administered approximately 15 min before radiotracer injection in blocking scans (Figure 3, bottom row). After PF-367, tissue concentration (summed 5–20 min) of [^11^C]OCM-44 was decreased relative to baseline, but for [^18^F]OCM-50 was slightly increased. In each case, radiotracer *f*_P_ was higher in blocking scans compared to baseline, increasing from 3% to 7–10% for [^11^C]OCM-44 and from 0.6% to 1.4% and 1.5% for [^18^F]OCM-50.

[^11^C]OCM-44 *V*_T_ was lower in blocking scans compared to baseline at the 0.03 mg/kg and 0.1 mg/kg dose, but was higher at the 0.25 mg/kg dose (Table 2 and Figure 5A). As a result, values of *V*_T_/*f*_P_ were calculated in each scan to account for the increase in *f*_P_ during the blocking scans and found to display consistent decreases in blocking scans and a linear relationship in occupancy plots (Figure 5B). [^11^C]OCM-44 binding reduction derived from *V*_T_/*f*_P_ measurements was 68% at 0.03 mg/kg PF-367, 93% with 0.1 mg/kg, and 88% with 0.25 mg/kg. Similarly, [^18^F]OCM-50 *V*_T_ was not consistently lower compared to baseline in a blocking scan with 0.1 mg/kg PF-367 (Table 2 and Figure 5C). With correction for the increase in *f*_P_, estimated reduction in specific binding was 80%, but considerable regional variability remained (Figure 5D).

## 3. Discussion

This work evaluated two of the most promising PET radioligands for imaging of cerebral GSK-3, [^11^C]OCM-44 and [^18^F]OCM-50, in non-human primates. Full in vivo quantification was performed including arterial blood sampling, kinetic modeling, and blocking studies with the GSK-3 antagonist PF-367. [^11^C]OCM-44 showed widespread brain uptake; however, a number of both hydrophilic and lipophilic radio-metabolites were identified in plasma across the scan time. Together with increasing *V*_T_ with longer scan duration, these data are consistent with radio-metabolite accumulation in brain within the 120-min scan. [^18^F]OCM-50 entered the brain readily and *V*_T_ values were stable over increasing scan durations, albeit the *f*_P_ and, accordingly, *V*_T_, were low (*V*_T_ < 1 mL/cm^3^). 

GSK-3 is a potential therapeutic target for a range of applications, including Alzheimer’s disease and other neurodegenerative disorders as well as psychiatric diseases such as major depression and bipolar disorder. To date, all efforts to advance a GSK-3-based therapeutic for these disorders of high socioeconomic burden have been unsuccessful. A selective GSK-3 PET radiotracer capable of quantifying alterations in enzyme availability and occupancy of GSK-3 targeting pharmaceuticals in the living brain would be a valuable tool for drug discovery. Our laboratory recently developed [^11^C]OCM-44 [18] and [^18^F]OCM-50 [19] as the most promising PET radiotracers. Subsequently, radiotracers based on PF-367 (including OCM-44) were applied to measure GSK-3 binding in vitro in both a preclinical model of Alzheimer’s disease and in post-mortem human brain tissue, which provides preliminary evidence of intriguing disease- and sex-related differences in binding [20] and underscores the utility of these radiotracers in brain research. Despite persistent efforts [12,13,14,15,21], it has been a challenge to develop a radioligand suitable for in vivo brain imaging of GSK-3 in clinical research. The present work is a notable step forward in these efforts, with thorough characterization in non-human primates of [^11^C]OCM-44 and [^18^F]OCM-50, the first GSK-3 radioligands to date demonstrated to enter the brain and show kinetics amenable to quantification. 

[^11^C]OCM-44 entered the brain readily and had *V*_T_ values in the range of 1.5–2.9 mL/cm^3^ with uptake throughout grey matter, broadly consistent with GSK-3 gene expression patterns in human brain [22]. However, *V*_T_ values were not stable over increasing scan duration and could not be reliably quantified using data from the full 120-min scan. Together with the emergence of a number of radio-metabolites in plasma beginning at early time points post-injection, this may suggest the presence of confounding brain-penetrant metabolites. We recently identified similar metabolic instability of [^11^C]OCM-44 in a mouse model [23] as well as preliminary evidence of radio-metabolites in mouse brain (unpublished observations). While the character of these metabolites is not known, brain-penetrant metabolites would adversely impact the quantitative analysis of [^11^C]OCM-44, with the accumulating radio-metabolite signal affecting the accuracy of kinetic model parameter estimates. Though its kinetics in brain may allow for accurate quantification from short scan durations, early emergence of radio-metabolites in plasma suggests that *V*_T_ may be overestimated even with truncated scan data in these analyses. If similar patterns occur in humans, [^11^C]OCM-44 may not be suitable for absolute quantification of brain GSK-3. Future preclinical studies including infusion scans may be useful to confirm and characterize metabolite uptake in brain, and further experiments to characterize metabolites may help inform future radiotracer design efforts. 

In contrast, [^18^F]OCM-50 *V*_T_ values were low but did not show a consistent increase with scan duration, with values derived from 90 min of scan data comparable to those from the full 120-min scan, though it should be noted that only one baseline scan was completed with full quantification. No evidence was observed of defluorination, judged by an absence of apparent bone uptake across scan time. Despite its higher in vitro affinity for GSK-3 [18], [^18^F]OCM-50 showed notably lower uptake and binding in the brain compared to [^11^C]OCM-44. This may reflect its lower *f*_P_ (<1% of radiotracer free in plasma). [^18^F]OCM-50 *K*_1_ values were also low throughout the brain, consistent with its low *f*_p_. In addition, the regional distribution of [^18^F]OCM-50 *V*_T_ values was not entirely consistent with that of [^11^C]OCM-44 (Table 2 and Appendix A). The modest correspondence between the two radiotracers and higher variability in regional blocking of [^18^F]OCM-50 (Figure 5C,D) suggests that [^18^F]OCM-50 may show some degree of off-target binding in vivo, though higher uncertainty in parameter estimates may also contribute to this variability. 

In blocking PET studies with administration of the selective GSK-3 inhibitor PF-367, tissue concentration (Figure 3, bottom row) and distribution volume (Table 2) were not consistently reduced for either radiotracer. However, these comparisons are complicated by higher *f*_P_ during blocking scans. Plasma analysis found *f*_P_ values approximately 3-fold higher in blocking scans, a change that was consistent across both radiotracers and each of the four animals. This results in increased amounts of radiotracer free to enter the brain in blocking scans, which may mask reduced binding site availability and result in higher tissue concentration in the blocking scan (Figure 3, bottom right). The cause of this increase is not clear—it is possible that a shared peripheral (plasma) binding site is blocked by the competitor drug, though saturation of such a binding site by the competitor is not expected. When accounting for this increase by using *V*_T_/*f*_P_ as an outcome measure for occupancy analyses (Figure 5B,D), a binding reduction of 68–93% is seen, consistent with a substantial reduction in binding site availability. PF-367 produced an 93% reduction in [^11^C]OCM-44 binding at a dose of 0.1 mg/kg in one animal and an 88% reduction at a dose of 0.25 mg/kg in the second animal, suggesting that selective binding at GSK-3 represents a large proportion of the [^11^C]OCM-44 signal. Based on occupancy plots using *f*_p_-normalized *V*_T_, estimated *V*_ND_/*f*_p_ for [^11^C]OCM-44 was 11.5 ± 4.8 mL/cm^3^ and for [^18^F]OCM-50 was 30.9 mL/cm^3^, giving *V*_ND_ estimates of 0.36 ± 0.15 mL/cm^3^ and 0.19 mL/cm^3^, respectively. From this, binding potential (*BP*_ND_) values are estimated to be 4.9 ± 0.6 (mean in cortical regions: 5.4) and 4.9 ± 0.8 (mean in cortical regions: 5.4) in the two animals for [^11^C]OCM-44, and 2.4 ± 0.9 (mean in cortical regions: 2.9) for [^18^F]OCM-50. However, because the role of *f*_P_ changes in these effects is not entirely clear, the degree of specific and nonspecific binding cannot be determined with confidence and quantitative comparisons should be interpreted with caution. Further PF-367 blocking studies performed in a single animal and using other doses within or near the same range (0.03–0.25 mg/kg) may be helpful to better understand the dose-response relationship and confirm if the blocking effect is saturated at ~90% binding reduction. Future studies may also choose to evaluate different competitor drugs to explore *f*_P_ variation effects and confirm in vivo target selectivity. Values and stability of *f*_P_ should be carefully evaluated in humans in considering these radiotracers for clinical translation.

## 4. Materials and Methods

### 4.1. Study Design

A total of five PET scans were performed with [^11^C]OCM-44 and four scans with [^18^F]OCM-44 in adult rhesus macaques. For [^11^C]OCM-44, two baseline and three blocking scans were acquired in two animals using 0.03 mg/kg, 0.1 mg/kg, and 0.25 mg/kg of PF-367 to assess a potential dose–response relationship. For [^18^F]OCM-50, baseline and pre-blocking scans with PF-367 at the intermediate dose, 0.1 mg/kg, were acquired in two different animals. Arterial sampling was performed for all scans, with the exception of one [^18^F]OCM-50 baseline scan where arterial line placement failed.

### 4.2. Radiochemistry

The synthesis of [^11^C]OCM-44 [18] and [^18^F]OCM-50 [19] was carried out as previously described with minor modifications. 

*[^11^C]OCM-44*: [^11^C]OCM-44 was produced using the AutoLoop (Bioscan, Washington, DC, USA) [24]. The desmethyl precursor (0.8–1.0 mg) in 80 µL of anhydrous DMF and 1.5 µL KOtBu was loaded into the loop and then ^11^C-methyl iodide was swept and trapped in the precursor solution at 18 mL/min. After a 5 min reaction at room temperature, the mixture was diluted with the HPLC mobile phase (62% 0.1 M ammonium formate with 0.5% AcOH, pH= 4.2 and 38%MeCN) and injected into a HPLC column (Phenomenex Luna C18 (2), 10 µm, 10 × 250 mm) for purification (flow rate of 5 mL/min). The radioactive peak at 13–14 min was collected, diluted with 50 mL of water, and passed through a C-18 Sep-Pak. The product was then eluted off the Sep-Pak with 1 mL of U.S. Pharmacopeial Convention (USP) absolute ethanol, followed by 3 mL of USP saline, into an empty syringe barrel and passed through a sterile membrane filter (0.22 µm) and collected in a sterile vial pre-charged with 7 mL of USP saline and 40 µL of 4.2% USP Na_2_CO_3_ solution to afford a formulated solution ready for injection. 

*[^18^F]OCM-50*: [^18^F]OCM-50 was synthesized from the ammonium triflate salt precursor. Cyclotron-produced ^18^F-fluoride was first trapped on the PS-HCO_3_^-^ cartridge, then eluted off with the solution of triethylammonium bicarbonate (TEAB, 2 mg) in 1 mL MeOH and azeotropically dried. The precursor solution (1.5–2.5 mg in 300 µL of dimethyl sulfoxide) was added and the mixture heated at 120 °C for 10 min. The crude reaction solution was diluted with the HPLC mobile phase (55% 0.1 M ammonium formate with 0.5% HOAc, pH = 4.2 and 45% MeCN) and passed through an alumina N cartridge to remove the ^18^F-fluoride, followed by purification with semi-preparative HPLC (xBridge column, 5 µm, 250 × 10 mm eluting with the aforementioned mobile phase at a flow rate of 5.0 mL/min). Post-processing and formulation procedures were the same as those for [^11^C]OCM-44. 

### 4.3. PET Imaging

Prior to each scan, animals were sedated with a combination of alfaxalone (2 mg/kg), midazolam (0.3 mg/kg), and dexmedetomidine (0.01 mg/kg) and maintained in an anesthetized state with 1.5–2.5% isoflurane. Heart rate, blood pressure, respiration rate, and oxygen saturation were monitored continuously. A catheter was placed in the radial artery for blood sampling. All PET scans were performed on a Focus 220 scanner (Siemens Medical Solutions, Knoxville, TN). An 8.5-min transmission scan was collected first for attenuation correction, followed by bolus injection of radiotracer in 10 mL over 3 min by an infusion pump (PHD 22/2000; Harvard Apparatus). Dynamic PET data were acquired in list mode for 120 min starting concurrent with the beginning of the injection, then binned into frames of increasing durations (6 × 30 s, 3 × 1 min, 2 × 2 min, 22 × 5 min). For pre-blocking scans, PF-367 (0.03, 0.01, or 0.25 mg/kg) was administered as a 5-min IV slow bolus beginning 15 min prior to radiotracer injection, then a 120-min scan was acquired as described.

### 4.4. Plasma Radiometabolite Analysis and Input Function Measurement

Arterial blood samples were collected manually at intervals post-injection to measure plasma input functions. Additional samples were collected at 3, 8, 15, 30, 60, and 90 min after tracer injection for metabolite analysis. Samples were processed and analyzed for unmetabolized parent fraction using an automatic column-switching high-performance liquid chromatography (HPLC) system [25]. Radio-metabolite-corrected arterial input functions were calculated as the product of the total plasma curve and the parent radiotracer fraction curve. The *f*_P_ for each scan was measured in triplicate using the ultrafiltration method. Briefly, the tracer solution was added to 3.0 mL of whole blood. After 10 min incubation at ambient temperature, the sample was centrifuged at 2930× *g* for 5 min and supernatant plasma (0.3 mL) was loaded onto the reservoir of a micropartition device (MilliporeSigma Centrifree Ultrafiltration, Burlington, MA) and centrifuged at 1228× *g* for 20 min. The *f*_P_ value was calculated as the ratio of radioactivity in the filtrate to that in the plasma. 

### 4.5. Image Processing

PET emission data were attenuation-corrected using the transmission scan then reconstructed using a Fourier rebinning and filtered back-projection algorithm. PET images summed from the first 10 min of each scan were registered to the animal’s anatomical magnetic resonance image, which had been registered to a brain atlas. Inverted transformations were applied to register the atlas region of interest (ROI) mask to the PET image and TACs were extracted from 14 brain ROIs: frontal, occipital, temporal, cingulate, and insular cortex, caudate, putamen, pallidum, amygdala, hippocampus, thalamus, pons, and cerebellum.

### 4.6. Kinetic Modeling and Blocking Study Analysis

Regional TACs and radio-metabolite-corrected arterial input functions were fitted with the 1TCM and 2TCM to generate estimates of regional *V*_T_ and *K*_1_. Quality of fit for each method was assessed visually and compared between models using the AIC. Precision of *V*_T_ estimates was assessed using rSE. Guo plots were constructed comparing [^11^C]OCM-44 and [^18^F]OCM-50 *V*_T_ values across brain regions in order to evaluate binding distribution and relative affinity of the two radiotracers [26]. For blocking scans, occupancy plots were constructed using the difference in *V*_T_ between baseline and blocking scans relative to baseline *V*_T_ values across ROIs. Changes in *V*_T_/*f*_p_ were also assessed given an observed difference in *f*_p_ between baseline and blocking scans (see Section 3). From the resulting linear relationship across ROIs, percent occupancy was determined to assess target selectivity of each radiotracer. 

## 5. Conclusions

[^11^C]OCM-44 and [^18^F]OCM-50, two of the candidate GSK-3 radioligands with the highest in vitro selectivity identified to date, readily entered the non-human primate brain and showed uptake throughout grey matter. Of the two radiotracers, [^11^C]OCM-44 had higher *f*_P_, higher *V*_T_ values and binding/blocking patterns more consistent with GSK-3 selectivity. However, the possibility of brain-penetrant radio-metabolites may hinder quantitative analysis for clinical applications, and changes in *f*_P_ require further study. Further work to assess test-retest variability of this compound would also be informative to evaluate translational potential. Nevertheless, characterization of these brain-penetrant radiotracers with selective binding and kinetics amenable to quantification represents an important step forward in the effort to develop a selective GSK-3 radiotracer suitable for use in brain imaging. Future work to improve metabolic stability, reduce plasma protein binding, and increase brain uptake may be fruitful.

## Figures and Tables

**Figure 1 pharmaceuticals-16-00194-f001:**
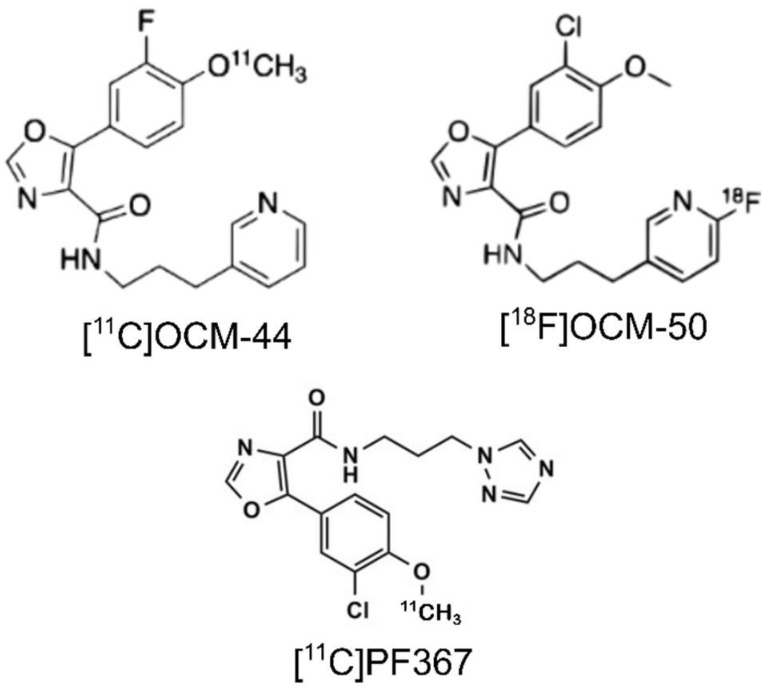
Chemical structures of [^11^C]OCM-44, [^18^F]OCM-50, and [^11^C]PF-367.

**Figure 2 pharmaceuticals-16-00194-f002:**
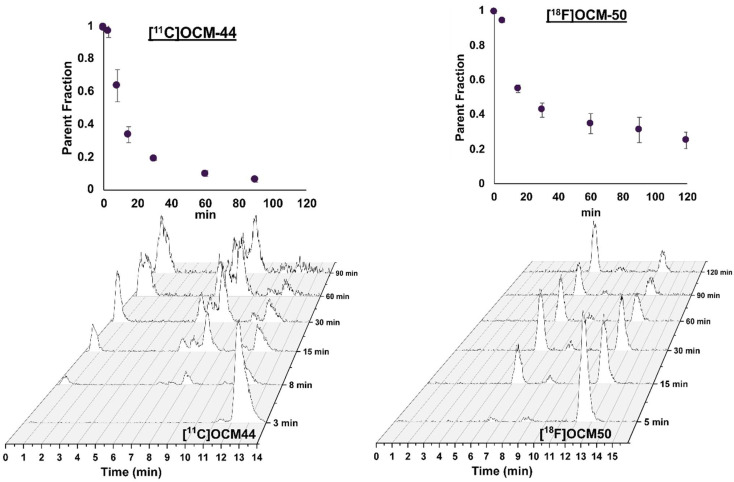
Parent fraction curves (top, mean ± S.D. parent fraction at each time point post injection) and plasma HPLC chromatograms (bottom) for baseline scans of [^11^C]OCM-44 and [^18^F]OCM-50 in non-human primates.

**Figure 3 pharmaceuticals-16-00194-f003:**
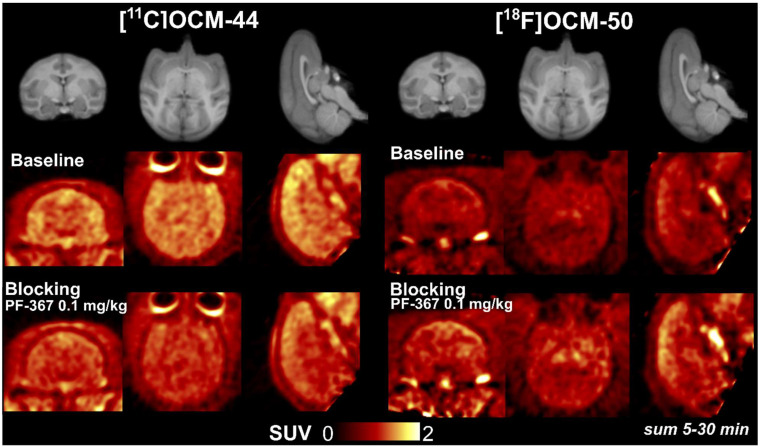
Brain uptake of [^11^C]OCM-44 and [^18^F]OCM-50 in rhesus macaque. Top row, anatomical MRI image of slice locations in template monkey brain; middle and bottom rows, template-registered baseline and PF-367 pre-blocking scans (summed SUV images, 5–20 min post-injection) performed in animal 2 (left) and animal 3 (right).

**Figure 4 pharmaceuticals-16-00194-f004:**
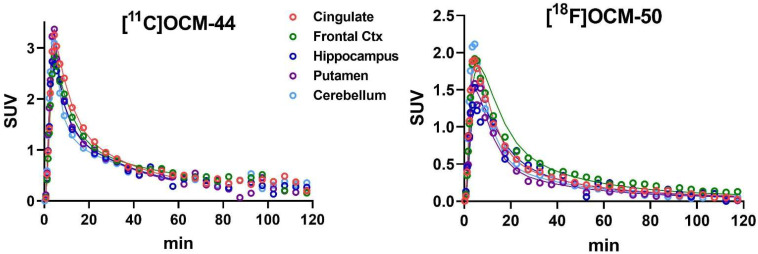
Time activity curves (TACs) for [^11^C]OCM-44 and [^18^F]OCM-50 baseline scans in non-human primates. Plots show TACs for representative brain regions along with preferred kinetic model fits (solid line): 2TCM fitted to the first 60 min of data for [^11^C]OCM-44 (**left**) and 1TCM fitted to the full 120 min of data for [^18^F]OCM-50 (**right**). Note the difference in scale.

**Figure 5 pharmaceuticals-16-00194-f005:**
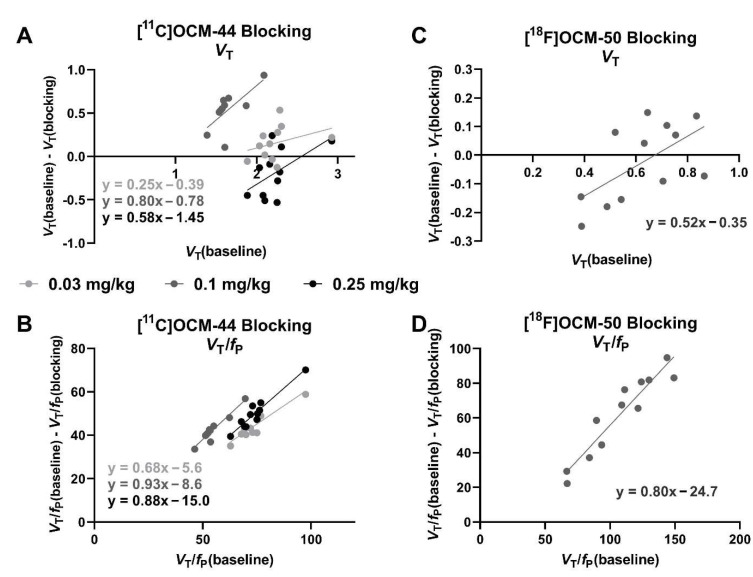
[^11^C]OCM-44 and [^18^F]OCM-50 blocking studies in non-human primates. Occupancy plots showing change in [^11^C]OCM-44 (**A**,**B**) and [^18^F]OCM-50 (C-D) *V*_T_ after pre-treatment with the GSK-3 antagonist PF-367 in each ROI using *V*_T_ values (**A**,**C**) or *V*_T_/*f*_P_ values to correct for higher *f*_P_ in blocking scans (**B**,**D**; see Results).

**Table 1 pharmaceuticals-16-00194-t001:** Injection parameters of [^11^C]OCM-44 and [^18^F]OCM-50 in four adult rhesus macaques.

	[^11^C]OCM-44		[^18^F]OCM-50	
	Baseline *n = 2*	Blocking *n = 3*	Baseline *n = 2*	Blocking *n = 2*
Drug (dose)	n/a	PF-367 (0.03, 0.1, or 0.25 mg/kg)	n/a	PF-367 (0.1 mg/kg)
Injected dose (MBq)	180, 174	177 ± 2.3	163, 171	155, 167
Injected mass (ng/kg)	29.6, 44.7	51.9 ± 28	5.1, 6.1	4.2, 14.3

**Table 2 pharmaceuticals-16-00194-t002:** *V*_T_ values for [^11^C]OCM-44 and [^18^F]OCM-50 in rhesus macaques.

	[^11^C]OCM-44 *V*_T_ (mL/cm^3^*)*					[^18^F]OCM-50 *V*_T_ (mL/cm^3^)		
	Baseline		Blocking *(PF-367)*			Baseline	Blocking *(PF-367)*	
	Animal 1	Animal 2	0.03 mg/kg (Animal 1)	0.1 mg/kg (Animal 2)	0.25 mg/kg (Animal 1)	Animal 3	0.1 mg/kg (Animal 3)	0.1 mg/kg (Animal 4)
Frontal cortex	2.60	2.09	2.34	1.15	4.43	0.87	0.94	0.96
Temporal cortex	2.10	1.68	2.08	7.33	2.61	0.72	0.44	0.62
Occipital cortex	2.26	1.57	1.98	1.02	2.54	0.84	0.70	0.79
Cingulate	2.92	1.87	2.71	1.29	2.74	0.71	0.80	1.01
Insula	2.25	1.65	2.38	0.98	2.78	0.39	0.53	0.68
Caudate	2.31	1.54	1.96	1.03	2.19	0.39	0.64	0.76
Putamen	2.16	1.61	2.01	1.01	2.25	0.54	0.70	0.84
Pallidum	2.19	1.39	2.23	1.14	1.95	0.49	0.67	0.78
Hippocampus	2.29	1.61	1.75	1.50	2.47	0.65	0.50	0.58
Amygdala	1.87	1.71	1.71	n.d.	3.40	0.45	0.19	0.37
Thalamus	2.03	1.56	1.91	1.03	2.16	0.52	0.44	0.71
Pons	1.88	1.48	1.94	2.31	2.34	0.75	0.68	0.76
Cerebellum	2.08	1.59	1.84	0.94	2.53	0.63	0.59	0.69

[^11^C]OCM-44 values derived from a 2TCM model fit to the first 60 min of data; [^18^F]OCM-50 values derived from a 1TCM fit to 120 min of data (see Section 3). n.d., not determined.

## Data Availability

Data is contained within the article and Appendix A.

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
