# Peer review of "In Vivo Imaging and Kinetic Modeling of Novel Glycogen Synthase Kinase-3 Radiotracers [11C]OCM-44 and [18F]OCM-50 in Non-Human Primates"

_pharmaceuticals, 2023, doi:10.3390/ph16020194_

Round 1

Reviewer 1 Report

  The development brain penetrating PET radiotracers for glycogen synthase kinase-3 (GSK-3) has gained increasing importance in the recent years. There are some evidences that dysregulation of GSK-3 is implicated in numerous psychiatric and neurological disorders for which GSK-3 inhibitors are being considered as therapeutic strategies. Early detection of the dysregulation of GSK-3 could and the ability to quantify GSK-3/ligand interactions can be used to understand different target engagement by the GSK-3 inhibitors in development of new drugs and evaluations of the dosage. In the previous studies the authors of the submitted manuscript have suggested several GSK3 radioligands labelled with 11C (T1/2 20.4 min), however none of them seemed to be suitable to further translation into human studies. In this manuscript the authors focused on the new GSK-3 targeting radiotracers based on the oxazole-4-carboxamide structural scaffold of the PF-367 as one of the most potent and selective GSK-3 inhibitors. Considering all the logistic short-coming of carbon-11 the 18F-labelled analogue was also developed and evaluated. Both radiotracers entitled [11C]OCM-44 and [18F]OCM-50 were evaluated into the human primates combining typical base-line and pre-blocking with PF-367 protocols. Despite both radiotracers exhibited sufficient penetration onto the brain, suitable kinetics and high selectivity to GSK-3, the arterial plasma radioHPLC analysis revealed formation of radiolabelled metabolites. In case of [11C]OCM-44 only 20% of the parent compound was found at 30 min after injection. The slower but still substantial metabolism occurred for [18F]OCM-50. From that point of view the results are not so optimistic to be translated into humans, however they worth publishing now. The results are worth considering as the important contribution into the field representing new data and insights for further developments of this series of the GSK-3 radioligands.  

There were some issues and questions that should be addressed prior to publication

1.      The authors mentioned that only the modest correspondence between the two radiotracers was observed. Interestingly, [11C]OCM-44 contains already fluorine atom in the scaffold. Why do not chose this position for 18F-labeling to get two identical radioligands labelled with different PET radioisotopes?

2.      Why different dosages of PF-367 were applied in the blocking studies for each radioligand?

3.      Are the any ideas about the nature of the metabolites arise from [18F]OCM-50? Did authors observe defluorination?

4.      On the Fig.1 there is a kind of mistake - [18F]PF-367 has no 18F label, probably that was [11C]PF-367.          

5.      Radiochemistry part is described in details, however the data on the radiochemical yield and synthesis are needed, even the authors referred to their previously published data.

6.      Line 101 - “triethylamine bicarbonate (TEAB, 2 mg)”, may be “triethylammonium bicarbonate”?

In conclusion, I would recommend the manuscript for the publication after answering the above questions.

Reviewer 2 Report

This paper presents a thorough evaluation of two new PET radiotracers for imaging of glycogen synthase kinase 3 (GSK-3) which may be a very promising target for treatment of various neurodegenerative disorders. The kinetic properties of the two tracers is studied in non-human primates using dynamic PET imaging including and experimental characterization of tracer metabolism and distribution in blood plasma. Also the specificity of the radiotracers are demonstrated in a displacement study after administration of a GSK-3 inhibitor. Unfortunately the image interpretation and accurate quantitation is challenged by properties of the radiotracers like metabolic instability, low free faction in blood plasma, and moderate uptake in the brain.

The manuscript is very well written and the methodological approach as well as the obtained results are clearly presented.

The work is recommended for publication after minor revision.

Specific comments:

1) Abstract, line 23: “VT” should be fully expressed when first mentioned (volume of distribution)

2) Abstract and figure 5, panel A and C: Use subscript for “T” in “VT”
